# Morphological and Phylogenetic Analyses Reveal Three New Species of Entomopathogenic Fungi Belonging to Clavicipitaceae (Hypocreales, Ascomycota)

**DOI:** 10.3390/jof10060423

**Published:** 2024-06-14

**Authors:** Zhi-Qin Wang, Jin-Mei Ma, Zhi-Li Yang, Jing Zhao, Zhi-Yong Yu, Jian-Hong Li, Hong Yu

**Affiliations:** 1Yunnan Herbal Laboratory, College of Ecology and Environmental Sciences, Yunnan University, Kunming 661500, China; w18314560773@163.com (Z.-Q.W.); 12022230164@mail.ynu.edu.cn (J.-M.M.); 18287972774@163.com (Z.-L.Y.); zhaojing@mail.ynu.edu.cn (J.Z.); 2The International Joint Research Center for Sustainable Utilization of Cordyceps Bioresources in China and Southeast Asia, Yunnan University, Kunming 661500, China; 3Yunnan Jinping Fenshuiling National Nature Reserve, Honghe 661500, China; ynfsljp@163.com (Z.-Y.Y.); 15808802076@163.com (J.-H.L.)

**Keywords:** *Moelleriella*, *Conoideocrella*, phylogenetic analyses, morphology, new taxa

## Abstract

This study aims to report three new species of *Conoideocrella* and *Moelleriella* from Yunnan Province, Southwestern China. Species of *Conoideocrella* and *Moelleriella* parasitize scale insects (Coccidae and Lecaniidae, Hemiptera) and whiteflies (Aleyrodidae, Hemiptera). Based on the phylogenetic analyses of the three-gene nrLSU, *tef-1α*, and *rpb1*, it showed one new record species (*Conoideocrella tenuis*) and one new species (*Conoideocrella fenshuilingensis* sp. nov.) in the genus *Conoideocrella*, and two new species, i.e., *Moelleriella longzhuensis* sp. nov. and *Moelleriella jinuoana* sp. nov. in the genus *Moelleriella*. The three new species were each clustered into separate clades that distinguished themselves from one another. All of them were distinguishable from their allied species based on their morphology. Morphological descriptions, illustrations, and comparisons of the allied taxa of the four species are provided in the present paper. In addition, calculations of intraspecific and interspecific genetic distances were performed for *Moelleriella* and *Conoideocrella*.

## 1. Introduction

In the family Clavicipitaceae, there are some genera that have been found to parasitize scale insects or whiteflies, such as *Aschersonia* Mont., *Dussiella* Pat., *Helicocollum* Luangsaard, Mongkols., Noisrip. & Thanak, *Hyperdermium* J.F. White, R.F. Sullivan, Bills & Hywel-Jones, *Hypocrella* Sacc., *Moelleriella* Bres., *Orbiocrella* D. Johnson, G.H. Sung, Hywel-Jones & Spatafora, *Regiocrella* Chaverri & K.T. Hodge, and *Samuelsia* P. Chaverri & K.T. Hodge [1,2,3,4,5]. Among them, *Aschersonia*, *Hypocrella*, and *Moelleriella* are relatively old genera and have more species. A taxonomic revision was undertaken by Chaverri et al. (2008) for the species formerly belonging to *Hypocrella* Sacc. s. l. (anamorph *Aschersonia* Mont. s. l.). They used three-gene phylogenetic analyses and morphological characters to classify *Hypocrella* Sacc. s. l. into *Hypocrella* s. str. (anamorph *Aschersonia*), *Moelleriella* (anamorph aschersonia-like), and *Samuelsia* (anamorph aschersonia-like) [3]. *Moelleriella* was distinguished from *Hypocrella* s. str. and *Samuelsia* by the fact that their ascospores can be disarticulated at the septa within the ascus, whereas those of the latter two cannot [3].

The genus *Moelleriella* was erected by Bresadola in 1896 to accommodate the type species *M. sulphurea*, which is currently regarded as a synonym of *M. phyllogena* (Mont.) P. Chaverri & K.T. Hodge (basionym *Hypocrella phyllogena* (Mont.) Speg.) [3]. There are currently 65 species in this genus, all of which have brightly colored stromata; obpyriform to subglobose perithecia; cylindrical asci; filiform multiseptate ascospores that disarticulate at the septa inside the ascus; and aschersonia-like anamorphs with fusoid conidia (according to the Index Fungorum, which is available online at http://www.indexfungorum.org; accessed on 11 December 2023 [6]). Among them, there are 30 species from the New World and 35 species from the Old World (according to the Index Fungorum, which is available online at http://www.indexfungorum.org; accessed on 29 August 2023 [3,6,7,8,9,10,11,12,13,14,15]). In China, four new species have been reported. Two species were reported in Yunnan Province, and another two were reported in Fujian Province [11,13,14,15]. In addition to these new published species, *Moelleriella* has also been distributed in other provinces in China. However, previous studies have primarily focused on morphology, with few molecular data available in public databases [16,17,18,19]. 

*Torrubiella* Boud. species infect a wide range of arthropods, mainly spiders and scale insects [4,20]. The genus currently has about 80 records in the index (according to the Index Fungorum, which is available online at http://www.indexfungorum.org; accessed on 11 December 2023). However, the fact that the previous studies identified species on the basis of their morphological characteristics resulted in a lack of molecular data for most species of *Torrubiella*. With the advent of molecular technology and the application of multigene phylogenetic analyses, species identification methods based on phylogenetic analyses combined with morphological characteristics have gradually gained recognition. Johnson et al. (2009) found that previous phylogenetic studies had shown that the genus *Torrubiella* was not monophyletic, but none of them had attempted to resolve this [4,21,22]. Subsequently, a multigene phylogenetic tree covering 10 species of *Torrubiella* was constructed by Johnson et al. to determine the phylogenetic position of these species [4]. Phylogenetic analyses showed that these species were distributed in Clavicipitaceae, Cordycipitaceae, and Ophiocordycipitaceae [4]. *Torrubiella tenuis* (Petch) D. Johnson, G.-H. Sung, Hywel-Jones & Spatafora and *Torrubiella luteorostrata* (Zimm.) D. Johnson, G.H. Sung, Hywel-Jones & Spatafora form a statistically well-supported clade in Clavicipitaceae. Therefore, a new genus, *Conoideocrella*, was proposed by Johnson et al. to accommodate the species *T. tenuis* and *T. luteorostrata*, and *T. luteorostrata* was designated as the type species [4]. The genus *Conoideocrella* was named thus for its perithecium with a conical shape that is similar to that of *Torrubiella* [4]. It currently contains three species, all of which have elongate, conical perithecia and planar stromata [23,24]. *Conoideocrella luteorostrata* was shown to be distributed in Seychelles, Sri Lanka, Java, Samoa, New Zealand, the far Eastern U.S.S.R., and Thailand [23,25]. *Conoideocrella tenuis* was known to be distributed in Sri Lanka and Thailand, and *C. krungchingensis* was known only to be in Thailand [23,24,25]. All three species have been reported to be able to parasitize scale insects [23,24].

Entomopathogenic fungi are widely distributed in China, and Yunnan Province is one of the richest provinces in China in terms of biodiversity. In this study, we collected some specimens with macro-morphological similarities to *Moelleriella* and *Conoideocrella* during an investigation of entomopathogenic fungi in Yunnan. A three-gene phylogenetic analysis revealed two new species of *Moelleriella*, one new species, and one known species of *Conoideocrella*. *Conoideocrella tenuis* is a recently newly recorded species in China.

## 2. Materials and Methods

### 2.1. Fungal Collection and Isolation

The specimens were collected from Bampo village, Jinuo Township, Jinghong City, and the Fenshuiling National Nature Reserve, Jinping County, Yunnan Province, China. In fields, whole leaves with stromata were collected, and some bark from branches with stromata was chipped off with a pocket knife. Then they were placed in sterilized plastic boxes and brought to the laboratory. The detailed procedure to obtain axenic cultures in this study was described in Yang et al. [15]. After the isolation of pure cultures, they were transplanted to PDA slant and grown for 10 days before being stored at 4 °C. The specimens were deposited in the Yunnan Herbal Herbarium (YHH) of Yunnan University, China. The strains were deposited at the Yunnan Fungal Culture Collection (YFCC) of Yunnan University, China.

### 2.2. Morphological Observations

Because of the small size of the stromata, a dissecting microscope (SZ61, Olympus Corporation, Tokyo, Japan) was used to observe their macro-morphological characteristics and measure them. The stromata were sectioned with a thickness of 30~40 µm for observations of their micro-morphological features using a HM525NX freezing microtome (Thermo Fisher Scientific, Waltham, MA, USA). The sections were placed on slides dripping with water or lactic acid in cotton blue. The observations were made and measurements were taken using a light microscope (Olympus BX53, Olympus Corporation, Tokyo, Japan). In order to observe and record the color and texture of the colonies, several new plates were transferred from the purified colonies and incubated in a 25 °C incubator for three weeks. The growth rate of colonies was used according to the method of Liu and Hodge [26] and was categorized as follows: fast growing (30–35 mm in diameter), moderately growing (20–30 mm in diameter), and slow growing (<20 mm in diameter).

### 2.3. DNA Extraction, PCR, and Sequencing

The specimens were washed with 75% alcohol, and the genomic DNA was extracted using the Genomic DNA Purification Kit (Qiagen GmbH, Hilden, Germany). The DNA of the cultures was extracted using cetyltrimethyl ammonium bromide (CTAB) following the procedure described by Liu et al. [26]. Three genes (nrLSU, *tef-1α*, and *rpb1*) were sequenced, and the following primer pairs were used for PCR amplification. LR5 (5′-ATCCTGAGGGAAACTTC-3′) and LR0R (5′-GTACCCGCTGAACTTAAGC-3′) were used to amplify the nuclear ribosomal large subunit (nrLSU) [27,28]. EF1α-EF (5′-GCTCCYGGHCAYCGTGAYTTYAT-3′) and EF1α-ER (5′-ATGACACCRACRGCRACRGTYTG-3′) were used to amplify the translation elongation factor 1α (*tef-1α*) [22,29]. RPB1-5′F (5′-CAYCCWGGYTTYATCAAGAA-3′) and RPB1-5′R (5′-CCNGCDATNTCRTTRTCCATRTA-3′) were used to amplify the largest subunits of RNA polymerase II (*rpb1*) [22,29]. The polymerase chain reaction (PCR) matrix and the PCR reactions were performed as described by Wang et al. [30]. A BIORAD T100TM thermal cycler (BIO-RAD Laboratories, Hercules, CA, USA) was used to perform amplification reactions. Then the PCR products were sequenced by the Beijing Genomics Institute (Chongqing, China).

### 2.4. Phylogenetic Analyses

Datasets of three genes (nrLSU, *tef-1α*, and *rpb1*) used to construct a phylogenetic tree were downloaded from GenBank and combined with the newly generated data in this study. The sequences downloaded were based on previous studies by Mongkolsamrit et al. [24] and Yang et al. [15]. Names, voucher information, and corresponding GenBank accession numbers of the taxa are listed in Table 1. Sequences were aligned, and poorly aligned regions were removed with MEGA v.6.06 [31]. The aligned three-gene sequences were concatenated using Phylosuite v1.2.2 [32]. Phylogenetic analyses were performed using BI and ML methods [33,34]. A maximum likelihood (ML) tree was created using IQ-tree v.2.1.3, and a Bayesian inference (BI) tree was created using MrBayes v.3.2.2 [35,36]. Modelfinder was used to select the best-fitting likelihood model [37]. The optimal model for the ML analyses was the TIM2+F+I+G4 model, with 5000 rapid bootstraps in a single run [38]. The optimal model for the BI analysis was the GTR+F+I+G4 model. The four Markov chain Monte Carlo simulations ran for 2 million generations from a random start tree with a sampling frequency of 100 generations. Twenty-five percent of initial sampled data were discarded as burn-in. Phylogenetic trees were viewed and edited in Figtree v.1.4.3 and visualized in Adobe Illustrator CS6. The interspecies and intraspecies genetic distances for the three genes (*tef-1α*, *rpb1*, and nrLSU) in *Moelleriella* and *Conoideocrella* were calculated using MEGAE v.6.06. Genetic distances were calculated by selecting the maximum composite likelihood model.

## 3. Results

### 3.1. Sequencing and Phylogenetic Analyses

Three-gene (nrLSU, *tef-1α*, and *rpb1*) sequences were generated from eight specimens and two living cultures (see Table 1). Three-gene sequences of 129 samples from 14 genera in the family Clavicipitaceae were used for the ML and BI phylogenetic analyses. *Pleurocordyceps aurantiaca* MFLUCC 17-2113 and *Pleurocordyceps marginaliradians* MFLU 17-1582 were used as the outgroups. The concatenated three-gene sequences contained 2726 bp (nrLSU: 935 bp, *tef-1α*: 1024 bp, and *rpb1*: 767 bp). Both the ML and BI analyses exhibited nearly consistent overall topologies. The results of the phylogenetic analysis showed three highly supported clades, viz., the Pulvinate clade (BP = 100%, PP = 1), the Globose clade (BP = 100%, PP = 1), and the Effuse clade (BP = 100%, PP = 1) (Figure 1). The Effuse clade was segregated into two sister clades, subclade I and subclade II. *Moelleriella* contains the Effuse clade and the Globose clade. Two new species of *Moelleriella* were distributed in the Effuse clade (*M. longzhuensis*) and the Globose clade (*M. jinuoana*). Three samples of *M. longzhuensis* were clustered closely with *M. rhombispora* (M. Liu & K.T. Hodge) M. Liu & P. Chaverri and formed a monophyletic clade in the Effuse clade with a high level of statistical support (BP = 100%, PP = 1). Three samples of *M. jinuoana* formed a monophyletic clade in the Globose clade with a high level of statistical support (BP = 91%, PP = 0.8). The genus *Conoideocrella* was clustered with *Orbiocrella* and had a new species (*C. fenshuilingensis*) and one known species (*C. tenuis*). Two samples of *C. fenshuilingensis* formed a monophyletic clade in *Conoideocrella* with a high level of statistical support (BP = 100%, PP = 1).

The genetic distances calculated based on the three genes (nrLSU, *tef-1α*, and *rpb1*) among interspecies and intraspecies in *Moelleriella* and *Conoideocrella* are shown in Appendix A. The intraspecific genetic distances for nrLSU, *tef-1α*, and *rpb1* in *Moelleriella* were 0–0.0276, 0–0.0428, and 0–0.0168, respectively. The interspecific genetic distances for nrLSU, *tef-1α*, and *rpb1* between the known species and *M. longzhuensis* were 0.03–0.08, 0.09–0.16, and 0.08–0.24, respectively, and those between the known species and *M. jinuoana* were 0.04–0.08, 0.10–0.15, and 0.15–0.25, respectively. In *Conoideocrella*, the intraspecific genetic distances of nrLSU, *tef-1α*, and *rpb1* were 0–0.0028, 0–0.0052, and 0.0019–0.0027, respectively, and the genetic distances between *C. fenshuilingensis* and the known species were 0.01–0.02, 0.06–0.09, and 0.08, respectively.
DICHOTOMOUS KEYS TO *CONOIDEOCRELLA* SPECIES1a.Stromata flattened pulvinate to discoid, planar, pulvinate, almost planar.............................................................................21b.Stromata scutate or hemi-globose....................................................................................................................*C. fenshuilingensis*2a.Perithecia < 600 µm long................................................................................................................................................................32b.Perithecia > 600 µm long.........................................................................................................................................*C. luteorostrata*3a.Stromata pale yellow, orange to reddish brown; Asci < 180 µm long; Conidia 8–15 × 2–4 μm..................................................................................................................................................................................................................................*C. krungchingensis*3b.Stromata white to orangish-pink; Asci > 180 µm long; Conidia 6.1–12.5 × 1.3–2.3 μm...............................................................................................................................................................................................................................................................*C. tenuis*
DICHOTOMOUS KEYS TO *MOELLERIELLA* SPECIESBased on teleomorphic characters1a.Part-ascospores > 16 µm long.……................................................................................................................................................21b.Part-ascospores < 16 µm long…..…...............................................................................................................................................32a.Perithecia embedded in stroma and scattered.............................................................................................................................42b.Perithecia embedded in top part of stroma, number perithecia per stroma > 30................................................*M. phyllogena*3a.Part-ascospores with rounded, blunt, or acute ends...................................................................................................................53b.Part-ascospores with truncated ends.................................................................................................................................*M. flava*4a.Part-ascospores ventricose cylindrical or curved with rounded ends and usually inflated in the middle; Part-ascospores< 20 µm long.......................................................................................................................................................................*M. basicystis*4b.Part-ascospores ventricose cylindrical or curved with rounded ends and usually inflated in the middle; Part-ascospores> 20 µm long.....................................................................................................................................................................*M. umbospora*5a.Stromata not thin-umbonate..........................................................................................................................................................65b.Stromata thin-umbonate, raised, with globose to subglobose base...............................................................*M. chiangmaiensis*6a.Part-ascospores 5–10 µm long........................................................................................................................................................76b.Part-ascospores 7–16 µm long........................................................................................................................................................87a.Stromata size < 2 mm.......................................................................................................................................................................97b.Stromata size > 2 mm.....................................................................................................................................................................108a.Part-ascospores < 3.5 µm width....................................................................................................................................................188b.Part-ascospores > 3.5 µm width.................................................................................................................................*M. colliculosa*9a.Stromata surface roughened..........................................................................................................................................*M. castanea*9b.Stromata surface not roughened..................................................................................................................................................1110a.Stromata suface tomentose...............................................................................................................................................*M. nivea*10b.Stromata suface smooth..............................................................................................................................................................1511a.Stromata yellowish white to white, pale yellow......................................................................................................................1211b.Stromata greyish yellow or reddish brown or dark brown almost black.............................................................................1312a.Stromata globose with head markedly constricted at base....................................................................................................1412b.Stromata pulvinate, base slightly constricted.....................................................................................................*M. zhongdongii*13a.Stromata greyish yellow and reddish brown...........................................................................................................*M. epiphylla*13b.Stromata dark brown almost black.......................................................................................................................*M. guaranitica*14a.Hypothallus present...................................................................................................................................................*M. disjuncta*14b.Hypothallus absent...................................................................................................................................................*M. boliviensis*15a.Stromata obconical.......................................................................................................................................................*.M. cornuta*15b.Stromata globose or subglose....................................................................................................................................................1616a.Stromata buff to pale greenish.............................................................................................................................*M. gaertneriana*16b.Stromata brownish orange, greyish brown, bark brown almost black.................................................................................1717a.Stromata subglose...........................................................................................................................................................*M. palmae*17b.Stromata globose............................................................................................................................................................*M. globosa*18a.Part-ascospores not ventricose...................................................................................................................................................1918b.Part-ascospores ventricose with rounded or acute ends...................................................................................*M. rhombispora*19a.Part-ascospores fusoid................................................................................................................................................................2019b.Part-ascospores cylindrical with round or blunt ends............................................................................................................2120a.Stromata surface tomentose.......................................................................................................................................................2220b.Stromata surface not tomentose................................................................................................................................................2321a.Perithecia embedded in stroma, scattered...............................................................................................................................2621b.Perithecia in gregarious but well-separated tubercles or gregarious tubercles..................................................................2722a.Perithecia in gregarious but well-separated tubercles.........................................................................................*M. simaoensis*22b.Perithecia embedded in stroma, scattered..............................................................................................................*M. puerensis*23a.Part-ascospores > 3 µm width...................................................................................................................................*M. turbinata*23b.Part-ascospores < 3 µm width...................................................................................................................................................2424a.Stromata yellowish white to white, pale yellow.....................................................................................................................2524b.Stromata yellow....................................................................................................................................................*M. macrostroma*25a.Hypothallus present.........................................................................................................................................................*M. libera*25b.Hypothallus absent........................................................................................................................................................*M. evansii*26a.Part-ascospores < 2 µm width......................................................................................................................................*M. sinensis*26b.Part-ascospores > 2 µm width....................................................................................................................................................2827a.Hypothallus present...................................................................................................................................................................2927b.Hypothallus absent.....................................................................................................................................................*M. nanensis*28a.Stromata thin pulvinate with pronounced tubercles or pulvinate with sloping side, tubercles half-embedded...............................................................................................................................................................................................................*M. sloaneae*28b.Stromata flat pulvinate.......................................................................................................................................*M. phukhiaoensis*29a.Stromata pulvinate with sloping sides or base slightly constricted.......................................................................*M. ochracea*29b.Stromata flat pulvinate...............................................................................................................................................................3030a.Stromata white, pale yellow to orange...........................................................................................................*M. chumphonensis*30b.Stromata white....................................................................................................................................................................*M. alba*
Based on anamorphic characters1a.Conidia > 4 µm width......................................................................................................................................................................21b.Conidia < 4 µm width......................................................................................................................................................................32a.Conidiomata number fewer than ten.............................................................................................................................................42b.Conidiomata number more than ten..............................................................................................................................................53a.Conidia < 6 µm long.........................................................................................................................................................................73b.Conidia > 6 µm long.........................................................................................................................................................................84a.Conidiomata locules simple depressions of surface without distinct rims.............................................................*M. epiphylla*4b.Conidiomata locules pezizoid......................................................................................................................................*M. turbinata*5a.Conidiomata scattered in stroma....................................................................................................................................*.M. globosa*5b.Conidiomata circular in stroma......................................................................................................................................................66a.Conidia ventricose with acute ends.............................................................................................................................*M. basicystis*6b.Conidia ventricose almost rhomboid with acute ends............................................................................................*M. umbospora*7a.Stromata greyish brown; thick pulvinate, obconical pulvinate.................................................................................*M. castanea*7b.Stromata pale yellow; discoid with distinct stud shape..................................................................................*M. pongdueatensis*8a.Conidiomata scattered or circular in stroma................................................................................................................................98b.Conidiomata arrangement in the central of stroma...................................................................................................................109a.Stromata flat pulvinate, dark orange to golden yellow.....................................................................................*M. phukhiaoensis*9b.Stromata not as above....................................................................................................................................................................1110a.Conidia < 16.5 µm long................................................................................................................................................................2410b.Conidia > 16.5 µm long......................................................................................................................................*M. chumphonensis*11a.Conidia > 3 µm width...................................................................................................................................................................1211b.Conidia < 3 µm width...................................................................................................................................................................1312a.Stromata dark brown, black.....................................................................................................................................*M. guaranitica*12b.Stromata yellowish white to white, pale yellow.....................................................................................................*M. phyllogena*13a.Conidia ventricose...................................................................................................................................................*M. rhombispora*13b.Conidia fusoid...............................................................................................................................................................................1414a.Conidia > 2 µm width...................................................................................................................................................................1514b.Conidia < 2 µm width...................................................................................................................................................................1615a.Conidiomata circular in stroma..................................................................................................................................*M. disjuncta*15b.Conidiomata scattered in stroma................................................................................................................................................1716a.Conidiomata circular in stroma........................................................................................................................................*M. libera*16b.Conidiomata scattered in stroma................................................................................................................................................2017a.Conidiomata fewer than ten....................................................................................................................................................... 1817b.Conidiomata more than ten.........................................................................................................................................................1918a.Stromata yellowish white to white, pale yellow...................................................................................................*M. madidiensis*18b.Stromata yellow, orange yellow to orange.................................................................................................................*M. jinuoana*19a.Stromata yellow......................................................................................................................................................*M. macrostroma*19b.Stromata yellowish white to white, pale yellow........................................................................................................*M. sloaneae*20a.Cultural on PDA compact, leathery............................................................................................................................*M. ochracea*20b.Cultural on PDA compact, floccose/tomentose........................................................................................................................2121a.Stromata yellowish white to white, pale yellow.......................................................................................................................2221b.Stromata brown..................................................................................................................................................*M. thanathonensis*22a.Stromata tuberculate, thick pulvinate, obconical pulvinate..............................................................................*M. zhongdongii*22b.Stromata flat or raised, globose to subglobose.........................................................................................................................2323a.Paraphyses present in conidioma....................................................................................................................*M. chiangmaiensis*23b.Paraphyses absent in conidioma................................................................................................................................*M. nanensis*24a.Paraphyses present in conidioma...............................................................................................................................................2524b.Paraphyses absent in conidioma................................................................................................................................................2625a.Stromata yellowish white to white, pale yellow.......................................................................................................................2725b.Stromata whitish to pale brown.......................................................................................................................*M. chumphonensis*26a.Conidiomata more than ten.........................................................................................................................................................2926b.Conidiomata fewer than ten........................................................................................................................................................3027a.Stromata thin pulvinate, almost effuse......................................................................................................................................2827b.Stromata flat to umbonate, globose to subglobose....................................................................................................*M. sinensis*28a.Conidia 8.8–14 × 1.6–3 μm........................................................................................................................................*M. simaoensis*28b.Conidia 9.7–13.4 × 1.3–2.3 μm...................................................................................................................................*M. puerensis*29a.Stromata scutate (a hemisphaerical central region abruptly attenuating and extending to the edge)....................................................................................................................................................................................................................................*M. evansii*29b.Stromata flat to umbonate, globose to subglobose...................................................................................*M. kanchanaburiensis*30a.Stromata yellowish white to white, whitish to pale yellow, pale yellow, yellow................................................................3130b.Stromata white.....................................................................................................................................................................*M. alba*31a.Conidia < 2 µm width..................................................................................................................................................................3231b.Conidia > 2 µm width...........................................................................................................................................*M. longzhuensis*32a.Stromata flat or raised, globose to subglobose; conidia 9–14 × 1–2 μm........................................................................*M. flava*32b.Stromata flat to umbonate, subglobose; conidia 7–10 × 1–2 μm...................................................................................*M. nivea*

### 3.2. Taxonomy

*Conoideocrella tenuis* (Petch) D. Johnson, G.H. Sung, Hywel-Jones & Spatafora, Mycol. Res. 113(3): 286 (2009), Figure 2.

≡*Torrubiella tenuis Petch*, Ann. Perad. 7, 323 (1923).

MycoBank No: MB 512029.

*Description*. Sexual morph: Teleomorphic stromata pulvinate, flattened pulvinate or almost planar, 2–4 mm in diam, white to orangish-pink, tomentose, rather loose internally, surrounded by a broad, fibrillose margin or hypothallus. Perithecia mostly distributed at the margin of the stroma or on the hypothallus, scattered or clustered, color deepens from the bottom to the top, white to pale brown, covered with hyphae up to two-thirds their height in mature perithecia, dozens of perithecia per stroma, elongated flask shape or elongated conic shape, 190–500 × 160–270 μm. Asci cylindrical, eight-spored, 190–480 × 3.3–5.5 μm, caps 2.5–3.5 μm thick. Ascospores whole, filiform, septate. Asexual morph: Not known.

*Culture characteristics*. Colonies grow slowly on PDA at 25 °C, attaining a diam of 15–17 mm in 21 days, greyish-white to cream-white mycelium at first, turning lilac with age. Colonies are loose on the surface and compact at the bottom. Hyphae smooth, septate, hyaline, 1.1–3.6 µm wide. Hirsutella-like asexual state arises from hyphae, conidiogenous structures with slender base tapering more or less evenly to a neck, hyaline, produced directly on hyphae of the stromatic colonies from ca. 5 wk onwards, 16.3–149.4 × 0.6–2.4 μm, and 0.3–1.3 μm wide at the apex. Conidia hyaline, smooth, fusiform and slightly curved, produced singly or in a group of two at the neck apex, 6.1–12.5 × 1.3–2.3 μm.

*Habitat*. Parasitic on *Aspidiotus destructor* on a jungle tree; on a black Aleyrodes on *Sarcococca pruniformis*; on a scale on *Hedyotis lessertianan*; on Aleyrodes on *Lasianthus walkerianus* and *Psychotria elongata*.

*Distribution*. Sri Lanka (type locality) and Thailand, China.

*Other material examined*. China, Yunnan Province, Jinghong City, Jinuo Township, Banpo village, 100°98′ E, 22°06′ N, alt. 1035 m, found on the underside of living leaves of dicotyledonous plants, 2 October 2022. Hong Yu and Zhi-Qin Wang (YHH CTBP221012, YHH CTBP221013, YHH CTBP22109315; YFCC CTBP22109316, living culture). Yunnan Province, Puwen Town, 100°97′ E, 22°52′ N, alt. 1020 m, found on the underside of living leaves of dicotyledonous plants, 3 August 2023, Hong Yu and Zhi-Qin Wang (YHH CTPW2308031; YHH CTPW23089310).

*Commentary*. The species *C. tenuis*, formerly in the genus *Torrubiella*, was reclassified by Johnson et al. [4] to *Conoideocrella*. In 1923, Petch described the morphological characteristics of *T. tenuis*, as well as its distribution sites and host insects. But there was no record of the size of the asci or ascospores. Hywel-Jones [23] collected specimens of *T. tenuis* in Thailand. They recorded the size of the asci and ascospores and isolated pure cultures. In our study, specimens of this species were collected in Yunnan, China, and it was found to be distributed in China. Its macromorphology and micromorphology were generally consistent with those described by Petch and Hywel-Jones and Evens, with one difference being that the materials used in this study extend the perithecium (190–900 × 160–270 μm) and asci (190–500 × 3.3–7.0 μm) size range of this species. Noteworthy, a hirsutella-like asexual state was observed on the stromatic colonies in the present study, which has not yet been observed in other studies. Unfortunately, as in the case of the Thai material, part-spores were not seen in the Chinese collection.

*Conoideocrella fenshuilingensis* Hong Yu bis, Z.L. Yang, Z.Q. Wang & J.M. Ma, sp. nov. Figure 3.

Mycobank No: 851868.

*Etymology*. Named after the Fenshuiling National Nature Reserve where the species was collected.

*Type*. China, Yunnan, Jinping County, the Fenshuiling National Nature Reserve. 103°49′ E, 22°82′ N, alt. 519 m, found on the underside of living leaves of dicotyledonous plants. 24 October 2023, Hong Yu (YHH CFFSL2310002, holotype).

*Description*. Sexual morph: Teleomorphic stromata scutate or hemi-globose, 3.0–3.4 mm in diam, snow-white, surrounded by a snow-white hypothallus. Perithecia densely distributed on stroma, a few distributed on hypothallus, scattered or clustered, color deepens from the bottom to the top, pale brown to black, dozens of perithecia per mature stroma, covered with hyphae up to two-thirds their height in mature perithecia, elongated flask shape or elongated conic shape, 259–795 × 144–242 μm. Asci cylindrical, eight-spored, 246–685 × 3.8–6.9 μm, caps 1.2–2.3 μm thick. Ascospores whole, filiform, septate. No anamorph was found with these stromata in nature. Asexual morph: Undetermined.

*Habitat*. Parasitic on scale insects (Coccidae, Sternorrhyncha, Hemiptera), found on the underside of living leaves of dicotyledonous plants.

*Distribution*. China, Yunnan Province, Jinping County.

*Other Material Examined*. China, Yunnan, Jinping County, the Fenshuiling National Nature Reserve. 103°49′ E, 22°82′ N, alt. 519 m, found on the underside of living leaves of dicotyledonous plants. 24 October 2023, Hong Yu and Zhi-Qin Wang (YHH CFFSL2310003, YHH CFFSL2310004, YHH CFFSL2310005, YHH CFFSL2310006, YHH CFFSL2310007, YHH CFFSL2310008, YHH CFFSL2310009, YHH CFFSL2310010, YHH CFFSL2310011, paratype).

*Commentary*. The phylogenetic analyses revealed that two samples of *C. fenshuilingensis* were grouped together and formed a separate clade in the genus *Conoideocrella*. Currently, only three species of *Conoideocrella* have been reported [4,24]. *Conoideocrella fenshuilingensis* was similar to the other three species in its elongated, conical perithecium and cylindrical asci. However, *C. fenshuilingensis* was particularly easy to distinguish from other species by the location of the perithecia on the stromata. Its perithecia was almost always grown on a hemispherical stroma, while that of *C. krungchingensis* was grown on a slight weft of hyphae, and that of *C. luteorostrata* was more commonly grown on the hypothallus [23,24]. Although the perithecium of *C. tenuis* was usually on the thicker part of the stroma, *C. fenshuilingensis* can be distinguished from *C. tenuis* by its hemispherical stroma, while *C. tenuis* has a flattened pulvinate or almost planar stroma [23].

*Moelleriella jinuoana* Hong Yu bis, Z.L. Yang, Z.Q. Wang & J.M. Ma, sp. nov. Figure 4.

Mycobank No: 851869.

*Etymology*. *Moelleriella jinuoana* was named after the Jinuo nationality, one of the 56 ethnic groups in China.

*Type*. China, Yunnan Province, Jinghong City, Jinuo Township, Banpo village, 100°98′ E, 22°06′ N, alt. 1046 m, found on trunks of dicotyledonous plants. 26 October 2023, Hong Yu (YHH MJBP2309031, holotype; YFCC MJBP23099451, ex-holotype living culture).

*Description*. Sexual morph: Not known. Asexual morph: Anamorphic stromata on natural substrate globose, surface smooth, yellow, orangish-yellow to orange, 1.2–2.3 mm diam, often with narrow hypothallus. Hyphae of stromata form compact textura epidermoidea. Conidiomata simple depressions of surface, producing grayish-yellow copious slime, several conidiomata per stroma, U-shaped or irregular shape, 110–373 × 181–286 μm. Phialides formed in a thick, compact palisade or cylindrical shape, 11.8–29.5 × 1–1.9 μm. Conidia unicellular, hyaline, smooth, fusoid with rounded ends, 8–11.5 × 2.1–2.9 μm. No praphyses were observed.

*Culture characteristics*. Colonies on PDA slow-growing, attaining a diam of 9–11 mm in 21 days at 25 °C. Stromatic colonies compact pulvinate, surface wrinkled, pale orange to orange. Conidial masses usually abundant, orange. Reverse of colony brownish.

*Habitat*. Parasitic on scale insects (Coccidae, Sternorrhyncha, Hemiptera) or whiteflies (Aleyrodidae, Sternorrhyncha, Hemiptera), on trunks of dicotyledonous plants.

*Distribution*. China, Yunnan Province, Jinghong City.

*Other Material examined*. China, Yunnan Province, Jinghong City, Jinuo Township, Bapo village, 100°98′ E, 22°06′ N, alt. 1046 m, found on trunks of dicotyledonous plants. 26 October 2023, Hong Yu and Zhi-Qin Wang (YHH MJBP2309032, paratype; YFCC MJBP23099452, ex-paratype living culture); Ibid., (YHH MJBP2309033, YHH MJBP2309034, YHH MJBP2309035).

*Commentary*. The three-gene phylogenetic analyses showed that the three samples of *M. jinuoana* were clustered in the Globe clade and closely related to *M. boliviensis* P. Chaverri & K.T. Hodge, *M. insperata* (Rombach, M. Liu, Humber, and K.T. Hodge) P. Chaverri & K.T. Hodge, and *M. macrostroma* (P. Chaverri and K.T. Hodge) P. Chaverri & K.T. Hodge. Morphologically, the stromata shape, texture, and color of *M. jinuoana* was significantly different from those of *M. boliviensis*, *M. insperata*, and *M. macrostroma* [3,39]. Ecologically, *M. jinuoana* was found on the trunk of dicotyledonous plants, while *M. boliviensis* and *M. insperata* were found on the underside of leaves, and *M. macrostroma* was found on the living vines of dicotyledonous plants [3,39].

*Moelleriella longzhuensis* Hong Yu bis, Z.L. Yang & Z.Q. Wang, sp. nov. Figure 5.

Mycobank No: 851870.

*Etymology*. Named after the Chinese name “Longzhu” of the plant *Dendrocalamus giganteus* Wall. ex Munro, to which the stromata were attached.

*Type*. China, Yunnan Province, Jinping County, the Fenshuiling National Nature Reserve, 103°22′ E, 22°78′ N, alt. 1436 m, found on the underside of the living leaves of *D. giganteus*. 22 October 2023, Hong Yu (YHH MLFSL2310012, holotype; YFCC MLFSL23109453, ex-holotype living culture).

*Description*. Sexual morph: Not known. Asexual morph: Anamorphic stromata with a subglobose head and constricted base (stud-shaped), white to pale yellow when immature becoming pale yellow to yellow when mature, 1–1.9 mm diam, surface pruinose or tomentose, with hypothallus 0.1–1.1 mm. Hyphae of stromata forming loose textura intricata to epidermoidea. Conidiomata simple depressions of surface, 319–481 × 237–448 μm, several conidiomata per stroma, but difficult to count because some of them are confluent with neighboring ones. Conidial masses pale yellow to yellow. In section, the Conidioma subglobose to globose. Phialides not observed. Conidia unicellular, hyaline, smooth, inflated at midpoint and tapering at both ends, 12–16.1 × 2.6–3.9 μm. No paraphyses observed.

*Culture characteristics*. Colonies on PDA slow-growing, attaining a diam of 6–7 mm in 21 days at 25 °C. Colonies pale yellow to pale yellowish brown, compact, forming a subglobose structure. Conidial masses usually abundant, usually forming several gushing bands, pale yellow to pale orange. Reverse of colony pale yellow to yellowish brown.

*Habitat*. Parasitic on scale insects (Coccidae, Sternorrhyncha, Hemiptera) and whiteflies (Aleyrodidae, Sternorrhyncha, Hemiptera), found on the underside of the living leaves of *D. giganteus*.

*Distribution*. China, Yunnan Province, Jinping County.

*Other material examined*. China, Yunnan Province, Jinping County, the Fenshuiling National Nature Reserve, 103°22′ E, 22°78′ N, alt. 1436 m, found on the underside of the living leaves of *D. giganteus*. 22 October 2023, Hong Yu and Zhi-Qin Wang (YHH MLFSL2310013, paratype; YFCC MLFSL23109454, ex-paratype living culture); Ibid., (YHH MLFSL2310014, YHH MLFSL2310015, YHH MLFSL2310016, YHH MLFSL2310017, YHH MLFSL2310018).

*Commentary*. Phylogenetically, the three samples of *M. longzhuensis* were grouped together in subclade II of the Effuse clade and formed a monophyletic group. It was sister to *M. rhombispora*, with a high level of statistical support in terms of the BI posterior probabilities (PP = 1) and the ML bootstrap proportions (BP = 100%). Morphologically, *M. longzhuensis* was similar to *M. rhombispora* in that its conidia were inflated at the midpoint and tapered at both ends [3]. However, *M. longzhuensis* exhibits significant morphological differences from *M. rhombispora* due to its color and the shape of its anamorphic stromata. The conidiomata of the former could be confluent with their neighbors but the those of the latter were not confluent. The former also had larger conidia (12–16.1 × 2.6–3.9 vs. 9–14 × 2.5–3 μm) and no paraphyses were observed [3].

## 4. Discussion

The calculation of genetic distances for the three genes showed that the interspecific genetic distances between the new species in this study and other species of the genus were greater than the intraspecific genetic distances (see Appendix A). This study resulted in the discovery of two new species of *Moelleriella* and one new species of *Conoideocrella* through phylogenetic analyses, morphological observations, and calculations of inter- and intraspecific genetic distances within genera.

The genus *Moelleriella* was divided into the Effuse clade and the Globose clade by Chaverri et al. [3]. The Effuse clade is composed of two sister clades, i.e., subclade I and subclade II [6]. Currently, subclade I includes 13 species, and subclade II includes 13 species [15]. The Effuse clade species were characterized as having effuse to thin, pulvinate stromata of loose hyphal tissue. Many species had hypothalli (e.g., *M. basicystis* P. Chaverri & K.T. Hodge; *M. disjuncta* (Seaver) P. Chaverri & K.T. Hodge; *M. libera* (Syd. and P. Syd.) P. Chaverri & M. Liu; *M. madidiensis* P. Chaverri & K.T. Hodge; *M. ochracea* (Massee) M. Liu & P. Chaverri; *M. phyllogena* (Mont.) P. Chaverri & K.T. Hodge; *M. pongdueatensis* Mongkols., Thanakitp. & Luangsa-ard; *M. raciborskii* (Zimm.) P. Chaverri, M. Liu & K.T. Hodge; *M. rhombispora* (M. Liu & K.T. Hodge) M. Liu & P. Chaverri; *M. thanathonensis* Y.P. Xiao, T.C. Wen & K.D. Hyde; *M. umbospora* P. Chaverri & K.T. Hodge; and *M. zhongdongii* (M. Liu & K.T. Hodge) M. Liu & P. Chaverri). These species had mostly whitish coloration and occasionally their stromata were pale yellow to orange (e.g., *M. basicystis*) [3,7,8,9,10,13,14,15]. The Globose clade species had globose and darker stromata, compact tissue, were hard or coriaceous, did not have hypothalli (except *M. sloaneae* (Pat.) P. Chaverri & K.T. Hodge), and had stromata that were usually 1–2 mm (except for *M. macrostroma* (P. Chaverri & K.T. Hodge) P. Chaverri & K.T. Hodge) [3]. However, these characteristics were not unique to each clade, and overlap could be found between the clades [3].

All three reported species of *Conoideocrella* were present in the field as telemorphic stromata with elongated flask-shaped or elongated conic-shaped perithecia [23,24]. In this way, it could be easily distinguished from *Moelleriella*, although it was in good agreement with *Moelleriella* in terms of growing environment and host category. Two genera, *Moelleriella* and *Conoideocrella*, were parasitic on whiteflies (Aleyrodidae, Hemiptera) or scale insects (Coccidae and Lecaniidae, Hemiptera) [3,23,24]. The fungus always completely obliterated the host, making it nearly impossible to identify the insect [3,23].

Scale insects and whiteflies are tiny, widely distributed parasites that suck plant sap, and many of these species are serious agricultural pests and act as vectors for viral plant diseases [40,41,42,43]. In the face of severe crop infestation by whiteflies, pesticides have been used mainly to suppress the whitefly population [44]. However, the overuse of pesticides has led to a certain degree of resistance to pesticides and harmful effects on non-target organisms and the environment [45]. The ability of *Moelleriella* and *Conoideocrella* species to parasitize large populations of whiteflies and scale insects in the wild gives these species the potential to be developed as green and non-polluting biological control agents. *Moelleriella libera* (anamorph *A. aleyrodis*) was the first species of *Moelleriella* to be applied to control whiteflies in Florida, U.S.A. [46]. Subsequently, there have been an increasing number of studies on the control of pests with *M. libera* [47,48,49,50,51]. Relatively few studies have been conducted on other species of *Molleriella* for biocontrol materials.

*Hypocrella* s. str. (anamorph *Aschersonia*) and *Molleriella* species are not only important biocontrol agents, but their metabolites also possess a wide range of biological activities, such as anti-tumor, anti-malarial, anti-bacterial, and insecticidal properties, and have great value within applications for biopesticides and pharmaceuticals [52,53,54,55]. Various compounds have been reported in studies of the metabolites of *Moelleriella*, such as benzophenones, terpenoids, cyclopeptides, steroids, and carotenoids [52,56,57,58,59,60]. In the studies of the metabolites of *Conoideocrella*, depsipeptides, bioxanthracenes and their monomers, conoideoxime A, hopane triterpenoid (zeorin), xanthone glycoside and naphthopyrone glycoside, phenolic glucosides, and chromane analogues have been reported [61,62,63,64,65,66,67]. In this paper, three new species of *Moelleriella* and *Conoideocrella* have been identified. This discovery provides valuable insights to facilitate further research, exploitation, and utilization of their metabolites and entomopathogenic fungi for effective biological control of scale insects or whiteflies.

## Figures and Tables

**Figure 1 jof-10-00423-f001:**
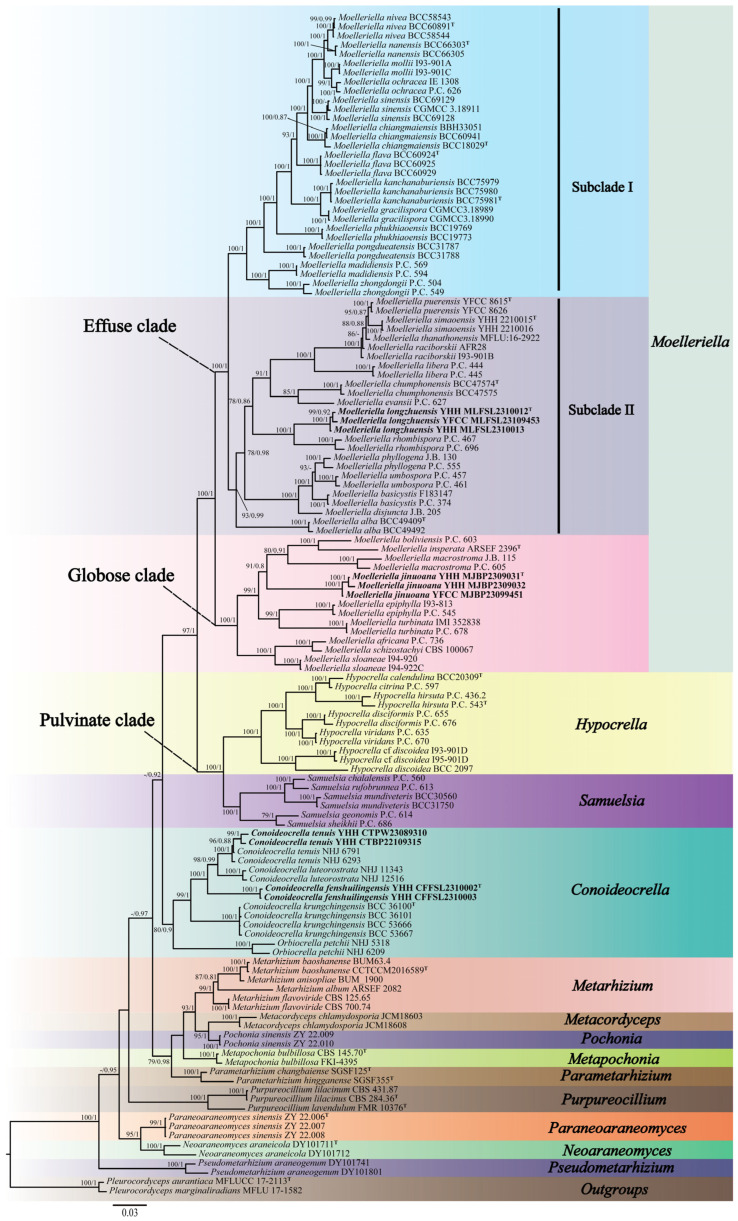
Phylogenetic relationships of 14 genera in Clavicipitaceae are based on the maximum likelihood (ML) and the Bayesian inference (BI) analyses using nrLSU, *tef-1α*, and *rpb1* sequences. Statistical support values greater than 70% are shown at the nodes for the BI posterior probabilities/the ML bootstrap proportions. The new taxa are highlighted in bold and ^T^ for ex-type material.

**Figure 2 jof-10-00423-f002:**
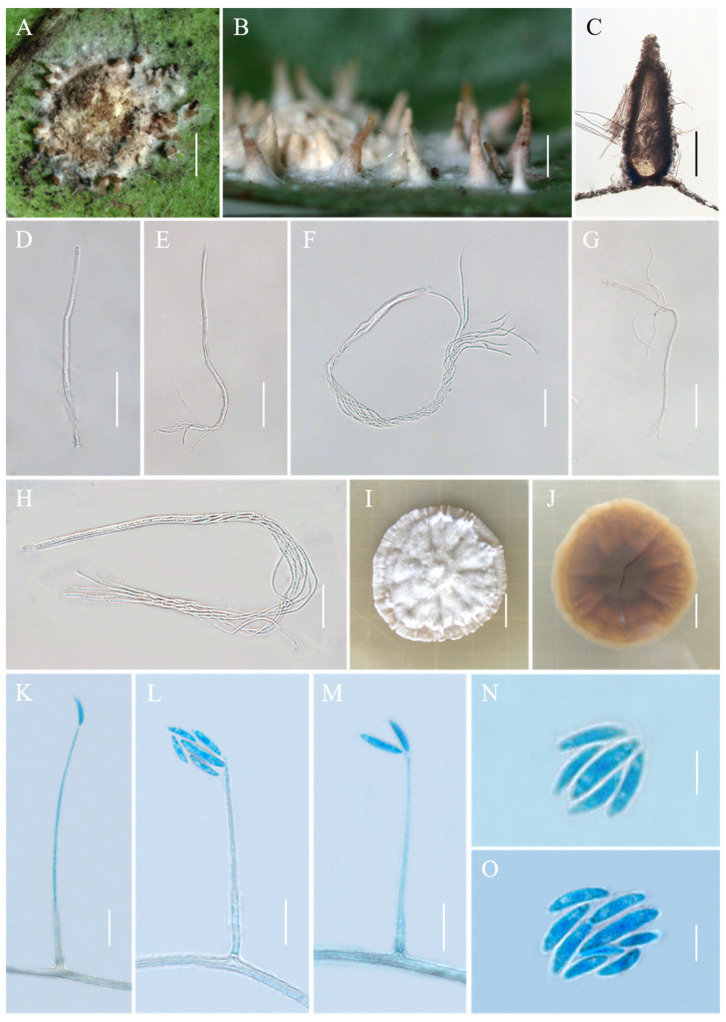
Morphology of *Conoideocrella tenuis*. (**A**,**B**) Telemorphic stroma containing perithecia; (**C**) Perithecium; (**D**–**H**) Mature asci with developing asci; (**I**) Obverse of colonies on PDA at 25 °C after 21 days; (**J**) Reverse of colonies on PDA at 25 °C after 21 days; (**K**–**O**) Conidia of hirsutella-like asexual stage on PDA. Scale bars: 1 mm (**A**,**B**); 200 µm (**C**); 50 µm (**D**,**E**); 25 µm (**F**); 50 µm (**G**); 20 µm (**H**); 1 cm (**I**,**J**); 10 µm (**K**–**M**); and 5 µm (**N**,**O**).

**Figure 3 jof-10-00423-f003:**
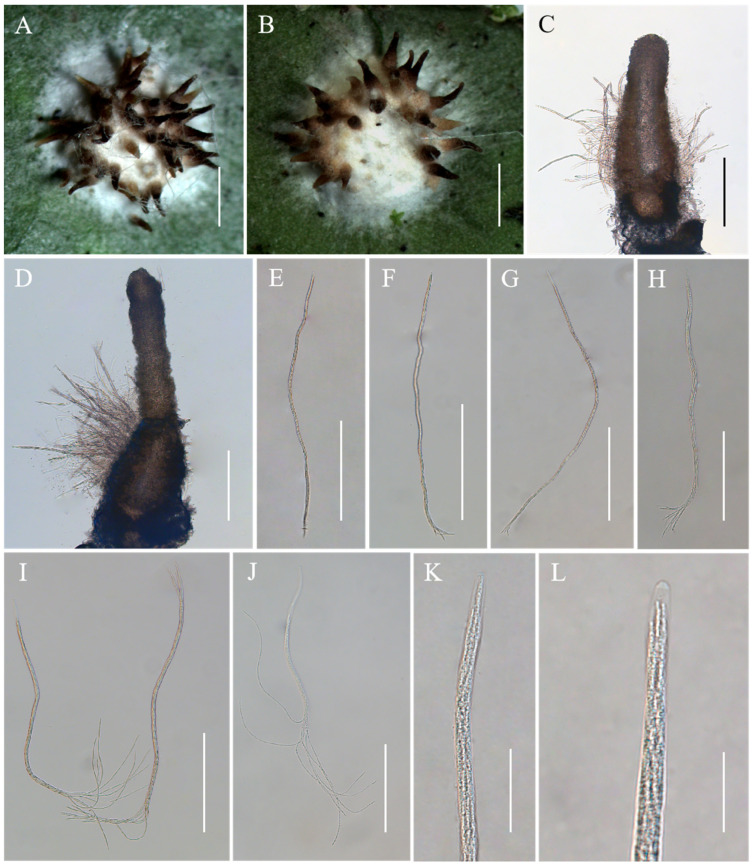
Morphology of *Conoideocrella fenshuilingensis*. (**A**,**B**) Telemorphic stroma containing perithecia; (**C**,**D**) Perithecium; (**E**–**L**) Mature asci with developing asci. Scale bars: 1 mm (**A**,**B**); 200 µm (**C**–**G**); 100 µm (**H**); 150 µm (**I**); 100 µm (**J**); 30 µm (**K**); and 20 µm (**L**).

**Figure 4 jof-10-00423-f004:**
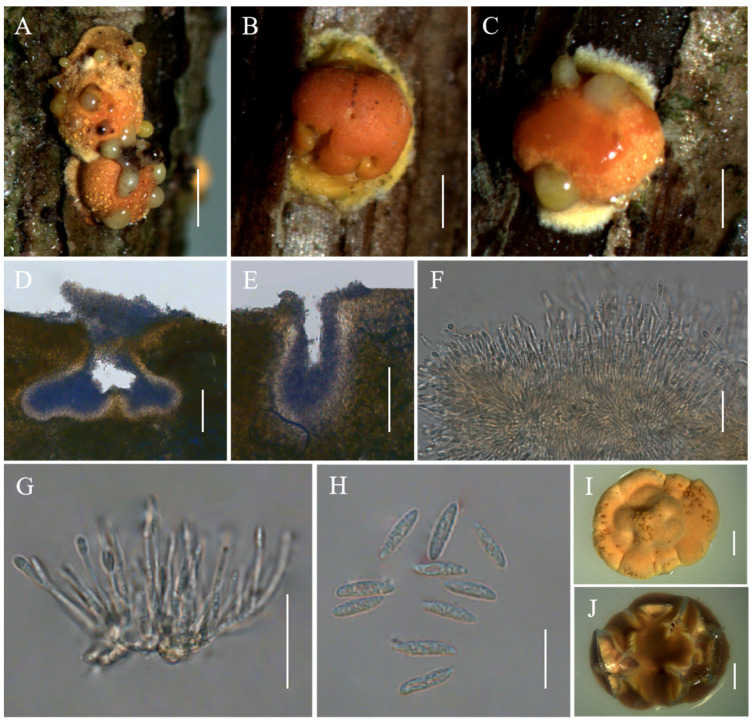
Morphology of *Moelleriella jinuoana*. (**A**–**C**) Anamorphic stromata containing conidiomata; (**D**,**E**) Section of stromata showing conidiomata; (**F**,**G**) Phialides with conidia at the tips; (**H**) Conidia; (**I**) Obverse of colonies on PDA at 25 °C after 21 days; (**J**) Reverse of colonies on PDA at 25 °C after 21 days. Scale bars: 1 mm (**A**); 0.5 mm (**B**,**C**); 100 µm (**D**,**E**); 20 µm (**F**,**G**); 10 µm (**H**); and 2 mm (**I**,**J**).

**Figure 5 jof-10-00423-f005:**
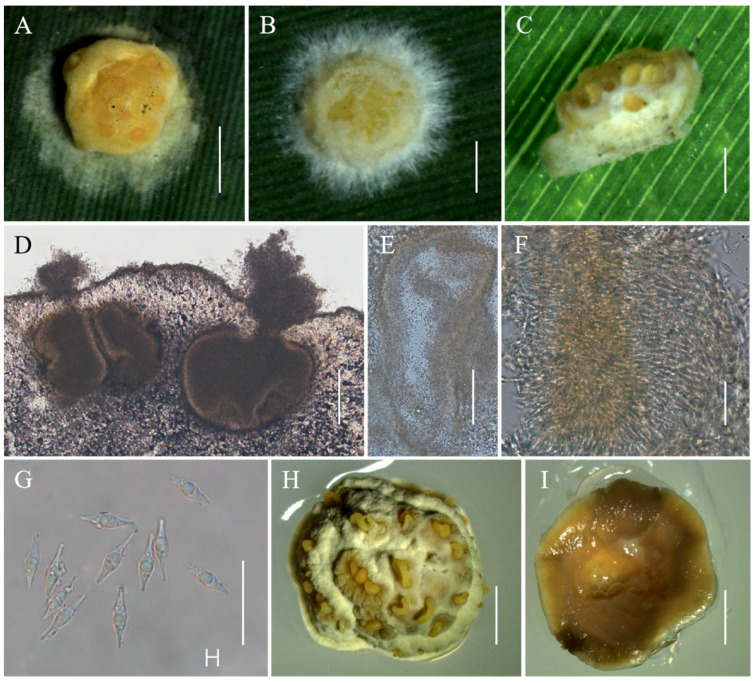
Morphology of *Moelleriella longzhuensis*. (**A**–**C**) Anamorphic stromata containing conidiomata; (**D**,**E**) Section of stromata showing conidiomata; (**F**) Phialides with conidia at the tips; (**G**) Conidia; (**H**) Obverse of colonies on PDA at 25 °C after 21 days; (**I**) Reverse of colonies on PDA at 25 °C after 21 days. Scale bars: 1 mm (**A**); 0.5 mm (**B**,**C**); 50 µm (**D**); 100 µm (**E**); 20 µm (**F**,**G**); and 2 mm (**H**,**I**).

**Table 1 jof-10-00423-t001:** Names, voucher information, and corresponding GenBank accession numbers of the taxa used in this study.

Species	Strain	Host	Origin	GenBank Accession Numbers		
				nrLSU	*tef-1α*	*rpb1*
** *Conoideocrella fenshuilingensis* **	**YHH CFFSL2310002 ^T^**	**Scale insects**	**China, Yunnan, Jinping County, the Fenshuiling National Nature Reserve**	**PP178583**	**PP177168**	**PP177158**
** *Conoideocrella fenshuilingensis* **	**YHH CFFSL2310003**	**Scale insects**	**China, Yunnan, Jinping County, the Fenshuiling National Nature Reserve**	**PP178584**	**PP177169**	**PP177159**
*Conoideocrella krungchingensis*	BCC 36100 ^T^	Scale insect	Thailand, Nakhon Si Thammarat Province, Khao Luang National Park	KJ435080	KJ435097	-
*Conoideocrella krungchingensis*	BCC 36101	Scale insect	Thailand, Nakhon Si Thammarat Province, Khao Luang National Park	KJ435081	KJ435098	-
*Conoideocrella krungchingensis*	BCC 53666	Scale insect	Thailand, Nakhon Si Thammarat Province, Khao Luang National Park	KJ435070	KJ435099	-
*Conoideocrella krungchingensis*	BCC 53667	Scale insect	Thailand, Nakhon Si Thammarat Province, Khao Luang National Park	KJ435071	KJ435100	-
*Conoideocrella luteorostrata*	NHJ 11343	Scale insect	-	EF468850	-	EF468906
*Conoideocrella luteorostrata*	NHJ 12516	Scale insect	-	EF468849	EF468800	EF468905
*Conoideocrella tenuis*	NHJ 6293	Scale insect	-	EU369044	EU369029	EU369068
*Conoideocrella tenuis*	NHJ 6791	Scale insect	-	EU369046	EU369028	EU369069
** *Conoideocrella tenuis* **	**YHH CTPW23089310**	**Scale insects**	**China, Yunnan, Jinping County, the Fenshuiling National Nature Reserve**	**PP178581**	**PP177166**	**PP177156**
** *Conoideocrella tenuis* **	**YFCC CTBP22109315**	**Scale insects**	**China, Yunnan, Jinping County, the Fenshuiling National Nature Reserve**	**PP178582**	**PP177167**	**PP177157**
*Hypocrella calendulina*	BCC 20309 ^T^	Scale insect nymph	Thailand, Ranong, Khlong Naka study trail, Khlong Naka Wildlife Sanctuary	GU552154	-	-
*Hypocrella citrina*	P.C. 597	Scale insects or whiteflies	Bolivia, Dpto. La Paz, San Jose de Uchipiamonas, Madidi National Park, Chalalan, trail to mountain along Eslabon River	AY986905	AY986930	-
*Hypocrella* cf *discoidea*	I93-901D	-	Côte D’Ivoire	EU392567	EU392646	EU392700
*Hypocrella* cf *discoidea*	I95-901D	-	Côte D’Ivoire	EU392568	EU392647	EU392701
*Hypocrella discoidea*	BCC 2097	-	Thailand	-	AY986945	DQ000346
*Hypocrella disciformis*	P.C. 655	Scale insects or whiteflies	Honduras, Dpto. Copan, Copan Ruinas, nature trail	EU392560	EU392643	EU392697
*Hypocrella disciformis*	P.C. 676	Scale insects or whiteflies	Honduras, Dpto. Copan, Copan Ruinas, nature trail	EU392566	EU392645	EU392699
*Hypocrella hirsuta*	P.C. 436.2	Scale insects or whiteflies	Mexico, Veracruz, Catemaco, town of Ejido Lopez-Mateo, project “Cielo, Tierra y Selva”, trail to the mountain	AY986922	AY986949	DQ000350
*Hypocrella hirsuta*	P.C. 543 ^T^	Scale insects or whiteflies	Bolivia, Dpto. La Paz, San Jose de Uchipiamonas, Madidi National Park, Chalalan, near Chalalan lodge	EU392569	EU392648	EU392702
*Hypocrella viridans*	P.C. 635	Scale insects or whiteflies	Honduras, Dpto. Atlantida, Tela, Lancetilla Natural Reserve	EU392572	EU392651	EU392705
*Hypocrella viridans*	P.C. 670	Scale insects or whiteflies	Honduras, Dpto. Copan, Santa Rita, Rio Amarillo, Peña Quemada Reserve	EU392574	EU392652	EU392706
*Metacordyceps chlamydosporia*	JCM18603	Soil under *Hibiscus rosa-sinensis*	Japan, Tokyo, Hachijo Island	-	AB758464	AB758667
*Metacordyceps chlamydosporia*	JCM18608	Soil	Japan, Hokkaido, Sapporo	-	AB758481	AB758684
*Metapochonia bulbillosa*	FKI-4395	Soil under *Q. serrata*	Japan, Nagano	AB709809	AB758460	AB758663
*Metapochonia bulbillosa*	CBS 145.70 ^T^	Root of *Picea abies*	Denmark	AF339542	EF468796	EF468902
*Metarhizium album*	ARSEF 2082	*Cofana spectra*	Sri Lanka	DQ518775	DQ522352	KJ398617
*Metarhizium anisopliae*	BUM_1900	Soil	China, Yunnan, Gaoligong Mountains	MH143820	MH143854	MH143869
*Metarhizium baoshanense*	CCTCCM 2016589 ^T^	Soil	China, Yunnan, Baoshan City, Taibao mountain	KY264174	KY264169	KY264180
*Metarhizium baoshanense*	BUM63.4	Soil	China, Yunnan, Baoshan City, Taibao mountain	KY264175	KY264170	KY264181
*Metarhizium flavoviride*	CBS 125.65	-	-	MT078854	MT078846	MT078862
*Metarhizium flavoviride*	CBS 700.74	-	-	MT078855	MT078847	MT078863
*Moelleriella africana*	P.C. 736	-	Ghana, Costa Rica, and Bolivia	AY986917	AY986943	DQ000344
*Moelleriella alba*	BCC49409 ^T^	Whitefly nymphs	Thailand, Narathiwat Province, Hala Bala Wildlife Sanctuary	JQ269646	KX254423	JQ256906
*Moelleriella alba*	BCC49492	Whitefly nymphs	Thailand, Narathiwat Province, Hala Bala Wildlife Sanctuary	JQ269645	KX254424	JQ256905
*Moelleriella boliviensis*	P.C. 603	Scale insects	Bolivia, Dpto. La Paz, San Jose de Uchipiamonas, Madidi National Park	AY986923	AY986950	DQ000351
*Moelleriella basicystis*	F183147 = CUP 067746	Scale insects and whiteflies	Panama, Chiriqui, Quebrada Hacha, San Juan Oriente, Besiko	EU392577	EU392653	-
*Moelleriella basicystis*	P.C. 374	Scale insects and whiteflies	Costa Rica, Guanacaste, Guanacaste Conservation Area, Rincon de la Vieja National Park	AY986903	AY986928	DQ000329
*Moelleriella chaiangmaiensis*	BCC18029 ^T^	Scale insect	Thailand, Chiang Mai, Doi Inthanon National Park	MT659360	MW091560	-
*Moelleriella chaiangmaiensis*	BBH33051	Scale insect	Thailand, Krabi, Jiranan Techaprasan’s house (Thap Prik)	MT659362	MT672277	MT672269
*Moelleriella chaiangmaiensis*	BCC60941	Scale insect	Thailand, Nakhon Ratchasima, Khao Yai National Park, Pha Kluaimai Waterfall Nature Trail	MT659361	MT672278	MT672270
*Moelleriella chumphonensis*	BCC47574 ^T^	Whitefly nymphs	Thailand, Chumphon, Phato Watershed Conservation and Management Unit	JQ269647	KX254421	JQ256907
*Moelleriella chumphonensis*	BBC47575	Whitefly nymphs	Thailand, Chumphon, Phato Watershed Conservation and Management Unit	JQ269648	KX254422	JQ256908
*Moelleriella disjuncta*	J.B.205	Scale insects and whiteflies	Panama, Fortuna, along trail 1 km west of STRI Biological Station	EU392578	EU392654	-
*Moelleriella epiphylla*	P.C. 545	Scale insects and whiteflies	Bolivia, Dpto. La Paz, San Jose de Uchipiamonas, Madidi National Park, Chalalan, trail Tapacare	EU392585	EU392660	EU392711
*Moelleriella epiphylla*	I93-813	Scale insects and whiteflies	Guyana, Matthew’s Ridge	EU392583	EU392656	EU392707
*Moelleriella evansii*	P.C. 627	Scale insects and whiteflies	Ecuador, Manabi. Y de la Laguna, Reserva Bilsa	AY986916	AY986942	DQ000343
*Moelleriella flava*	BCC60924 ^T^	Scale insects	Thailand, Nakhon Ratchasima, Khao Yai National Park, Mo Sing To Nature Trail	KF951146	KX254430	MT672271
*Moelleriella flava*	BCC60925	Scale insects	Thailand, Nakhon Ratchasima, Khao Yai National Park, Mo Sing To Nature Trail	KF951147	KX254431	MT672272
*Moelleriella flava*	BCC60929	Scale insects	Thailand, Nakhon Ratchasima, Khao Yai National Park, Mo Sing To Nature Trail	KX298238	KX254432	MT672273
*Moelleriella gracilispora*	CGMCC3.18989	Whitefly nymphs	China, Fujian Province, Wuyishan City, Wu Yi Mountain	KC964202	KC964191	KC964179
*Moelleriella gracilispora*	CGMCC3.18990	Whitefly nymphs	China, Fujian Province, Wuyishan City, Wu Yi Mountain	KC964203	KC964192	KC964180
*Moelleriella insperata*	ARSEF 2396 ^T^	-	Philippines, Los Baños, Laguna, NNE slope of Mount Makiling	AY518374	DQ070029	EU392713
** *Moelleriella jinuoana* **	**YHH MJBP2309031 ^T^**	**Scale insects and whiteflies**	**China, Yunnan Province, Jinghong City, Jinuo Township, Banpo village**	**PP178643**	**PP177170**	**PP177160**
** *Moelleriella jinuoana* **	**YHH MJBP2309032**	**Scale insects and whiteflies**	**China, Yunnan Province, Jinghong City, Jinuo Township, Banpo village**	**PP178644**	**PP177171**	**PP177161**
** *Moelleriella jinuoana* **	**YFCC MJBP23099451**	**Scale insects and whiteflies**	**China, Yunnan Province, Jinghong City, Jinuo Township, Banpo village**	**PP178645**	**PP177172**	**PP177162**
*Moelleriella kanchanaburiensis*	BCC75979	Scale insects	Thailand, Kanchanaburi, Takhian Thong Waterfall Nature Trail	MT659363	MT672279	MT843900
*Moelleriella kanchanaburiensis*	BCC75980	Scale insects	Thailand, Kanchanaburi, Takhian Thong Waterfall Nature Trail	MT659364	MT672280	MT843901
*Moelleriella kanchanaburiensis*	BCC75981 ^T^	Scale insects	Thailand, Kanchanaburi, Takhian Thong Waterfall Nature Trail	MT659365	MT672281	-
*Moelleriella libera*	P.C. 444	Scale insects and whiteflies	Mexico, Veracruz, Catemaco, Ejido Lopez Mateo town, project `Cielo, Tierra y Selva’, trail to mountain	EU392591	EU392662	EU392714
*Moelleriella libera*	P.C. 445	Scale insects and whiteflies	Mexico, Veracruz, Catemaco, Ejido Lopez Mateo town, project `Cielo, Tierra y Selva’, trail to mountain	AY986900	AY986925	DQ000326
** *Moelleriella longzhuensis* **	**YHH MLFSL2310012 ^T^**	**Scale insects and whiteflies**	**China, Yunnan Province, Jinping County, the Fenshuiling National Nature Reserve**	**PP178646**	**PP177173**	**PP177163**
** *Moelleriella longzhuensis* **	**YHH MLFSL2310013**	**Scale insects and whiteflies**	**China, Yunnan Province, Jinping County, the Fenshuiling National Nature Reserve**	**PP178647**	**PP177174**	**PP177164**
** *Moelleriella longzhuensis* **	**YFCC MLFSL23109453**	**Scale insects and whiteflies**	**China, Yunnan Province, Jinping County, the Fenshuiling National Nature Reserve**	**-**	**PP177175**	**PP177165**
*Moelleriella macrostroma*	J.B. 115	Scale insects	Costa Rica, Heredia, Sarapiquí, La Selva Biological Station	AY986920	AY986947	DQ000348
*Moelleriella macrostroma*	P.C. 605	Scale insects	Bolivia, La Paz Department, Province Franz Tamayo, San José de Uchipiamonas, Madidi National Park	AY986919	AY986946	DQ000347
*Moelleriella madidiensis*	P.C. 569	Scale insects	Bolivia, La Paz Department, Province Franz Tamayo, San José de Uchipiamonas, Madidi National Park	AY986915	AY986941	DQ000342
*Moelleriella madidiensis*	P.C. 594	Scale insects	Bolivia, La Paz Department, Province Franz Tamayo, San José de Uchipiamonas, Madidi National Park	EU392595	EU392666	EU392718
*Moelleriella mollii*	I93-901A	-	Côte D’Ivoire	EU392599	EU392667	EU392719
*Moelleriella mollii*	I93-901C	-	Côte D’Ivoire	EU392600	EU392668	EU392720
*Moelleriella nanensis*	BCC66303 ^T^	Scale insects	Thailand, Nan, Doi Mongkhon Nature Trail	KX298236	KX254427	MW085940
*Moelleriella nanensis*	BCC66305	Scale insects	Thailand, Nan, Doi Mongkhon Nature Trail	MW080317	KX254428	MW085941
*Moelleriella nivea*	BCC60891 ^T^	Scale insects	Thailand, Surat Thani, Khao Sok National Park, Wing Hin Waterfall Nature Trail	MW080318	MT672282	MW085942
*Moelleriella nivea*	BCC58543	Scale insects	Thailand, Surat Thani, Khao Sok National Park, Wing Hin Waterfall Nature Trail	MT659366	MT672283	MT672274
*Moelleriella nivea*	BCC58544	Scale insects	Thailand, Surat Thani, Khao Sok National Park, Wing Hin Waterfall Nature Trail	MT659367	MT672284	MT843898
*Moelleriella ochracea*	P.C. 626	Scale insects and whiteflies	Ecuador, Manabi. Y de la Laguna, Reserva Bilsa, primary forest	EU392604	EU392670	EU392722
*Moelleriella ochracea*	IE 1308 = P.C. 726	Scale insects and whiteflies	Mexico, Veracruz. Emiliano Zapata Municipality, Plan Chico	EU392601	EU392669	EU392721
*Moelleriella phukhiaoensis*	BCC19769	Scale insect nymphs	Thailand, Chaiyaphum Province, Bueng Pan Protect Forest Unit, Phu Khiao Wildlife Sanctuary	KT880502	-	KT880506
*Moelleriella phukhiaoensis*	BCC19773	Scale insect nymphs	Thailand, Chaiyaphum Province, Bueng Pan Protect Forest Unit, Phu Khiao Wildlife Sanctuary	KT880503	-	KT880507
*Moelleriella phyllogena*	P.C. 555	Scale insects and whiteflies	Bolivia, Dpto. La Paz, San Jose de Uchipiamonas, Madidi National Park, Chalalan, trail Tapacare	EU392610	EU392674	EU392726
*Moelleriella phyllogena*	J.B. 130	Scale insects and whiteflies	Panama, Fortuna, on leaf of Costa	EU392608	EU392672	EU392724
*Moelleriella pongdueatensis*	BCC31787	Scale insect nymphs	Thailand, Chiang Mai Province, Pong Dueat Pa Pae Geyser	KT880500	KX254433	KT880504
*Moelleriella pongdueatensis*	BCC31788	Scale insect nymphs	Thailand, Chiang Mai Province, Pong Dueat Pa Pae Geyser	KT880501	KX254434	KT880505
*Moelleriella puerensis*	YFCC 8615 ^T^	Whiteflies	China, Yunnan Province, Puer City, Simao District, Xinfang Reservoir	MW786748	MW815596	MW815595
*Moelleriella puerensis*	YFCC 8626	Whiteflies	China, Yunnan Province, Puer City, Simao District, Xinfang Reservoir	MW786750	MW815598	MW815594
*Moelleriella raciborskii*	AFR28	-	Ghana, Central Region, Jukua District, Kakum National Park, wet semideciduous forest	DQ070113	EU392675	EU392727
*Moelleriella raciborskii*	I93-901	-	Côte D’Ivoire	EU392611	EU392676	EU392728
*Moelleriella rhombispora*	P.C. 467	Scale insects and whiteflies	Costa Rica, Heredia, La Selva Biological Station, Camino Cantarrana	AY986908	AY986933	DQ000334
*Moelleriella rhombispora*	P.C. 696	Scale insects and whiteflies	Honduras, Yojoa, Los Pinos, Cerro Azul Meambar National Park	EU392618	EU392680	EU392732
*Moelleriella schizostachyi*	CBS 100067	-	Thailand	AY986921	AY986948	DQ000349
*Moelleriella sinensis*	BCC69128	Scale insects	Thailand, Chiang Mai, Doi Inthanon National Park, Mae Chaem Junction (KM.38) Nature Trail	KX298234	KX254425	MT843899
*Moelleriella sinensis*	BCC69129	Scale insects	Thailand, Chiang Mai, Doi Inthanon National Park, Mae Chaem Junction (KM.38) Nature Trail	KX298235	KX254426	MT672275
*Moelleriella sinensis*	CGMCC 3.18911	Whitefly nymphs	China, Fujian Province, Wu Yi Mountain	MK412091	-	MK412101
*Moelleriella simaoensis*	YHH 2210015 ^T^	Whiteflies	China, Yunnan Province, Puer City, Simao District, Xinfang Reservoir	OQ621807	OQ623179	OQ616915
*Moelleriella simaoensis*	YHH 2210016	Whiteflies	China, Yunnan Province, Puer City, Simao District, Xinfang Reservoir	OQ621808	OQ623180	OQ616916
*Moelleriella sloaneae*	I94-920	Scale insects or whiteflies	Guatemala, Tikal	EU392621	EU392682	EU392734
*Moelleriella sloaneae*	I94-922C	Scale insects or whiteflies	Belize, Cayo, Rio Frio	EU392622	EU392683	EU392735
*Moelleriella thanathonensis*	MFLU:16-2922	Unidentified insect	Thailand, Chiang Rai, Headquarter of Thanathon orchard	-	KY646200	-
*Moelleriella turbinata*	IMI 352838	Scale insects and whiteflies	Mexico, Estado Veracruz, Xalapa	EU392625	EU392685	EU392737
*Moelleriella turbinata*	P.C. 678	Scale insects and whiteflies	Honduras, Dpt. of Atlántida, Tela, Pico Bonito National Park	EU392627	EU392687	EU392739
*Moelleriella umbospora*	P.C. 461	Scale insects or whiteflies	Mexico, Chiapas, Palenque	EU392628	EU392688	EU392740
*Moelleriella umbospora*	P.C. 457	Scale insects or whiteflies	Mexico, Chiapas, Palenque	AY986904	AY986929	DQ000330
*Moelleriella zhongdongii*	P.C. 504	Scale insects or whiteflies	Costa Rica, Heredia, La Selva Biological Station, Sendero Cantarrana	EU392631	EU392689	EU392741
*Moelleriella zhongdongii*	P.C. 549	Scale insects or whiteflies	Bolivia, Yungas	EU392632	EU392690	EU392742
*Neoaraneomyces araneicola*	DY101711 ^T^	Spider	China, Guizhou, Duyun City, Qiannan Buyi and Miao Autonomous Prefecture	MW730609	MW753033	-
*Neoaraneomyces araneicola*	DY101712	Spider	China, Guizhou, Duyun City, Qiannan Buyi and Miao Autonomous Prefecture	MW730610	MW753034	-
*Orbiocrella petchii*	NHJ 5318	Scale insect	Thailand, Kaeng Krachan National Park, trail to Tor Tip Waterfall	EU369040	EU369021	EU369062
*Orbiocrella petchii*	NHJ 6209	Scale insect	Thailand	EU369039	EU369023	EU369061
*Parametarhizium changbaiense*	SGSF125 ^T^	On litter of forest	China, Jilin Province, Changbai Mountains	MN589994	MN908589	-
*Parametarhizium hingganense*	SGSF355 ^T^	On litter of forest	China, Heilongjiang Province, Greater Hinggan mountains	MN061635	MN065770	-
*Paraneoaraneomyces sinensis*	ZY 22.006 ^T^	Soil	China, Guizhou Province, Kaili City	OQ709260	OQ719626	**-**
*Paraneoaraneomyces sinensis*	ZY 22.007	Soil	China, Guizhou Province, Kaili City	OQ709261	OQ719627	**-**
*Paraneoaraneomyces sinensis*	ZY 22.008	Soil	China, Guizhou Province, Kaili City	OQ709262	OQ719628	**-**
*Pleurocordyceps aurantiaca*	MFLUCC 17-2113 ^T^	Larvae (Coleopteran)	Thailand, Prachuap Khiri Khan	MG136910	MG136875	MG136866
*Pleurocordyceps marginaliradians*	MFLU 17-1582	Cossidae larvae	Thailand, Chiang Mai, Te Mushroom Research Center	-	MG136878	MG136869
*Pochonia sinensis*	ZY 22.009	Soil	China, Guizhou Province, Kaili City	OQ709263	OQ719629	**-**
*Pochonia sinensis*	ZY 22.010	Soil	China, Guizhou Province, Kaili City	OQ709264	OQ719630	**-**
*Pseudometarbizium araneogenum*	DY101741	Spider	China, Guizhou, Duyun City, Qiannan Buyi and Miao Autonomous Prefecture	MW730618	MW753037	**-**
*Pseudometarbizium araneogenum*	DY101801 ^T^	Spider	China, Guizhou, Duyun City, Qiannan Buyi and Miao Autonomous Prefecture	MW730623	MW753039	**-**
*Purpureocillium lavendulum*	FMR 10376 ^T^	Soil	Venezuela, Caracas	FR775489	FR775516	FR775512
*Purpureocillium lilacinum*	CBS 284.36 ^T^	Soil	-	FR775484	FR734156	FR775507
*Purpureocillium lilacinum*	CBS 431.87	*Meloidogyne* sp.	-	EF468844	EF468791	EF468897
*Samuelsia geonomis*	P.C. 614	Scale insects or whiteflies	Bolivia, Dpt. of La Paz, San Jose de Uchipiamonas, Madidi National Park, Chalalan, trail to mountain along Eslabon River	EU392638	EU392692	EU392744
*Samuelsia sheikhii*	P.C. 686	Scale insects or whiteflies	Honduras, Yojoa, Los Pinos, Cerro Azul Meambar National Park	EU392639	EU392693	EU392745
*Samuelsia chalalensis*	P.C. 560	Whiteflies	Bolivia, Dpt. of La Paz, San Jose de Uchipiamonas, Madidi National Park	EU392637	EU392691	EU392743
*Samuelsia mundiveteris*	BCC40021	Scale insect nymphs	Thailand, Chiang Mai Province, Mae Chaem Junction (km 38), Doi Inthanon National Park	GU552152	GU552145	-
*Samuelsia mundiveteris*	BCC40022	Scale insect nymphs	Thailand, Chiang Mai Province, Mae Chaem Junction (km 38), Doi Inthanon National Park	GU552153	GU552146	-
*Samuelsia rufobrunnea*	P.C. 613	Insects	Bolivia, Dpt. of La Paz, San Jose de Uchipiamonas, Madidi National Park, Chalalan, trail to mountain along Eslabon River	AY986918	AY986944	-

New species are shown in bold. ^T^ ex-type material. ARSEF: ARS Collection of Entomopathogenic Fungal Cultures, New York, U.S.A; BBH: the BIOTEC Culture Herbarium; BCC: the BIOTEC Culture Collection; CBS: Centraalbureau voor Schimmelcultures Fungal Biodiversity Centre, Utrecht, the Netherlands; CCTCC: China Center for Type Culture Collection; CGMCC: the China General Microbiological Culture Collection Center; CUP: the Cornell University Plant Pathology Herbarium; FKI: Fungal Collection of Kitasato Institute for Life Sciences, Kitasato University, Tokyo, Japan; FMR: the culture collection of the Faculty of Medicine in Reus; JB: Joseph Bischoff, personal collection; JCM: Japan Collection of Microorganisms, Wako, Japan; MFLU: Mae Fah Luang University; MFLUCC: the Mae Fah Luang University Culture Collection; NHJ: Nigel Hywel-Jones, personal collection; P.C: Herbier Cryptogamique, Dépt. Systématique et Évolution, Muséum National d’Histoire Naturelle; YFCC: the Yunnan Fungal Culture Collection of Yunnan University, China; YHH: the Yunnan Herbal Herbarium of Yunnan University, China.

## Data Availability

The DNA sequence data obtained in this study have been deposited in GenBank. The accession numbers can be found in the article (Table 1).

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
