# Peer review of "Morphological and Phylogenetic Analyses Reveal Three New Species of Entomopathogenic Fungi Belonging to Clavicipitaceae (Hypocreales, Ascomycota)"

_jof, 2024, doi:10.3390/jof10060423_

Round 1

Reviewer 1 Report

The manuscript “Morphological and phylogenetic analyses reveal three new species of entomopathogenic fungi belonging to Clavicipitaceae (Hypocreales, Ascomycota)” presents the occurrence of new species of entomopathogenic fungi belonging to the genera Moelleriella and Conoideocrella.

Dear authors, the manuscript, in my opinion, is well written. The introduction is clear and presents what motivated the research, using updated and necessary literature. The material and methods topic presents the well-written sections “Fungal collection and isolation”, “Morphological observations”, “DNA extraction, PCR, and sequencing” and “Phylogenetic analyses”, methodologies that are widely used and described in the literature. The results are robust, following the proposed methodology. The discussion is adequate and based on the data found by the authors. With the aim of contributing to the improvement of the manuscript, I make some considerations for the authors’ evaluation:

L 46-47 and L55. Check for citations like “IndexFungorum: http://www. indexfungorum.org, August 29, 2023” as a reference, are in accordance with the journal’s standards.

L64 and L69. Complement the reference “Johnson et al.”

L95. I recommend that authors inform the number of days and not “several days”.

L 150 – 345. The topic results could be better presented. There is a lot of text that should be in the introduction or discussion. Example: L 306-314 “Commentary”.

L 345. The Coccidae and Aleyrodidae belong to the suborder Sternorrhyncha, order Hemiptera.

​

No additional comments

​

Author Response

Response to Reviewer 1 Comments

Point 1: L46-47 and L55. Check for citations like “IndexFungorum: http://www. indexfungorum.org, August 29, 2023” as a reference, are in accordance with the journal’s standards.

Response 1: We have read the Instructions for Authors to change it to (IndexFungorum. Available online: http://www. indexfungorum.org, accessed on 29 August 2023).

Point 2: L64 and L69. Complement the reference “Johnson et al.”

Response 2: Thanks very much for the reviewer’s suggestion. Added in the manuscript as requested by the reviewer.

Point 3: L95. I recommend that authors inform the number of days and not “several days”.

Response 3: Thanks very much for the reviewer’s suggestion. Changes have been made in the manuscript as requested by the reviewer.

Point 4: L 150 – 345. The topic results could be better presented. There is a lot of text that should be in the introduction or discussion. Example: L 306-314 “Commentary”.

Response 4: Thanks very much for the reviewer’s suggestion. The comments added to each new species after the description of the morphological features are intended to discuss the differences between this species and its close species, and we do not think that this part needs to be added to the introduction and discussion.

Point 5: L 345. The Coccidae and Aleyrodidae belong to the suborder Sternorrhyncha, order Hemiptera.

Response 5: Thanks very much for the reviewer’s suggestion. Changes have been made in the manuscript as requested by the reviewer.

Reviewer 2 Report

The paper provides a sound description of three new fungal species and adds to distribution of a known species of fungi belonging to the group of great scientific and practical importance. The methodology is adequate, including the most appropriate technics of multilocus phylogeny and histology. The results are well illustrated and mostly convincing.

However, the proof that the new isolates are separate taxa at the species level is concluded only in the phylogenetic tree without any indication of sequence identity levels between the new and previously described taxa to certify that the difference is profound enough.

Another problem is that the host species or genus is not defined. It may sometimes be problematic with the perished insects covered by fungal mycelia, but COI barcoding may give the clue with many taxa. If it not feasible to obtain this information, at least the species of dicotyledonous plants on which the scale bugs and the whiteflies were collected should be mentioned as an indirect indication of the potential host range and tritrophic associations of the fungus

Other comments are given below.

L25: I never heard of, and could find over the internet, the taxon Clavicipitidae. Seemingly, none of the papers cited refer to this term

L78-79: why using past tense? Has the situation described here changed over time and it is not so nowadays?

L83: similarly, why using past tense here? Is it not what you have just discovered?

L87-89: it doesn’t seem fair to introduce the new taxa names in the main text BEFORE the taxa are described

L96: is “herbal herbarium” not a tautology? Is it the official title of this collection?

L97: why only one strain was deposited?

L105-106: is it the method description or the protocol instruction?

L153: what does it mean “a sequence of 129 samples”? Is it a single or multiple sequences?

L155 and possible elsewhere: use italics for the Latin epithets of genus and species

L171-173: how relevant is this information for the goals of the study?

L228: what “part-spores” stand for?

L345: what the “back” stands for?

L390: what “host category” stands for?

L394-396: why using past tense here? What has changed, is it not so nowadays?

L400: “to host” doesn’t mean to parasitize, to infect etc. Plants may host insects, Insects may host fungi, but not vice versa

L407: it is unwise to start the new paragraph with a demonstrative pronoun which refers to an object above. Think of it as a new story so that the reader is not confused, since multiple taxa are mentioned in the previous paragraph.

L410: the reference here is in Chinese. Does it mean that you could not find an appropriate reference which would be available to a global audience?

L418: which data support the idea that the newly found fungi are useful for biocontrol of pests?

L419: what metabolites of pests are meant here?

L38: do not = could not

L46: inverted order of subject and predicate is not convenient for scientific texts

L51: When using formal written conventions, a conjunction, by definition, connects two clauses within a sentence. As such, a sentence cannot start with a conjunction when writing formally because there is no initial clause to connect to the second

L53: The genus is a taxonomic term, it is not an object which infects arthropod itself, better say “the representatives of the genus infect”

L54: “currently” doesn’t correspond to the past tense, it’s more likely the present

L129: After = Then

L145-146: the predicate is not obvious here

L161: sisterclades = sister clades

L271: a mixture of singular and plural forms

L285 and elsewhere: I guess several plants must have more than one trunk

L355: highly = high

L407: materials = agents

L410-413: starting two consequent phrases with identical sentences is a poor practice

Author Response

Response to Reviewer 2 Comments

Point 1: However, the proof that the new isolates are separate taxa at the species level is concluded only in the phylogenetic tree without any indication of sequence identity levels between the new and previously described taxa to certify that the difference is profound enough.

Response 1: Thanks very much for the reviewer’s suggestion. The identification of new species is often based on phylogenetic trees supplemented by morphology. At present, in Moelleriella and Conoideocrella, there is no relevant study of how much difference there is between the sequence of a species and the known species that can be divided into new species. Previous studies have been conducted by constructing phylogenetic trees, where a species can be classified as a new species if it can be distinguished from other known species and forms independent branches.

Point 2: Another problem is that the host species or genus is not defined. It may sometimes be problematic with the perished insects covered by fungal mycelia, but COI barcoding may give the clue with many taxa. If it not feasible to obtain this information, at least the species of dicotyledonous plants on which the scale bugs and the whiteflies were collected should be mentioned as an indirect indication of the potential host range and tritrophic associations of the fungus.

Response 2: Many thanks to the reviewer’s suggestion. We have tried to extract the DNA of the whole stroma and then amplified the COI, but because the host has almost been consumed, the amplification failed. The identification of plants has not been successful because we lack the relevant knowledge of plant taxonomy.

Point 3: L25: I never heard of, and could find over the internet, the taxon Clavicipitidae. Seemingly, none of the papers cited refer to this term.

Response 3: Many thanks to the reviewer’s suggestion. That's a misspelling on our part. It should be Clavicipitaceae here.

Point 4: L78-79: why using past tense? Has the situation described here changed over time and it is not so nowadays?

Response 4: Many thanks to the reviewer’s suggestion. We have changed the tense to the present tense in the manuscript.

Point 5: L83: similarly, why using past tense here? Is it not what you have just discovered?

Response 5: Many thanks to the reviewer’s suggestion. Because this result and the new species were discovered before the article was written, the past tense is used.

Point 6: L87-89: it doesn’t seem fair to introduce the new taxa names in the main text BEFORE the taxa are described.

Response 6: Thanks very much for the reviewer’s suggestion. Changes have been made in the manuscript as requested by the reviewer.

Point 7: L96: is “herbal herbarium” not a tautology? Is it the official title of this collection?

Response 7: Yes, it is the official title of this collection.

Point 8: L97: why only one strain was deposited?

Response 8: Thanks very much for the reviewer’s suggestion. It has been changed to plural in the manuscript.

Point 9: L105-106: is it the method description or the protocol instruction?

Response 9: This is the method description.

Point 10: L153: what does it mean “a sequence of 129 samples”? Is it a single or multiple sequences?

Response 10: This refers to multiple sequences, which we have modified in the manuscript.

Point 11: L155 and possible elsewhere: use italics for the Latin epithets of genus and species.

Response 11: Many thanks to the reviewer’s suggestion. Changes have been made in the manuscript as requested by the reviewer.

Point 12: L171-173: how relevant is this information for the goals of the study?

Response 12: Thanks very much for the reviewer’s suggestion. We have deleted it.

Point 13: L228: what “part-spores” stand for?

Response 13: The “part-spores” also called “secondary ascospores”, are formed by ascospores disarticulating at the septa. 

Point 14: L345: what the “back” stands for?

Response 14: We have changed the back to the underside of the living leaves, as detailed in the manuscript.

Point 15: L390: what “host category” stands for?

Response 15: This refers to the fact that both genera are parasitic on scale insects or whiteflies.

Point 16: L394-396: why using past tense here? What has changed, is it not so nowadays?

Response 16: Thanks very much for the reviewer’s suggestion. It has been changed to the present tense.

Point 17: L400: “to host” doesn’t mean to parasitize, to infect etc. Plants may host insects, Insects may host fungi, but not vice versa.

Response 17: Thanks very much for the reviewer’s suggestion. Changes have been made in the manuscript as requested by the reviewer.

Point 18: L407: it is unwise to start the new paragraph with a demonstrative pronoun which refers to an object above. Think of it as a new story so that the reader is not confused, since multiple taxa are mentioned in the previous paragraph.

Response 18: Thanks very much for the reviewer’s suggestion. Changes have been made in the manuscript as requested by the reviewer.

Point 19: L410: the reference here is in Chinese. Does it mean that you could not find an appropriate reference which would be available to a global audience?

Response 19: Thanks very much for the reviewer’s suggestion. We have added some literature to the manuscript.

Point 20: L418: which data support the idea that the newly found fungi are useful for biocontrol of pests?

Response 20: Thanks very much for the reviewer’s suggestion. Because the range of pests is so broad, and species of Moelleriella and Conoideocrella are parasitic on mealybugs or whiteflies, we have changed pests to scale insects or whiteflies in the manuscript.

Point 21: L419: what metabolites of pests are meant here?

Response 21: Thanks very much for the reviewer’s suggestion. The metabolites here refer to the metabolites of Moelleriella and Conoideocrella, perhaps because we did not express them clearly, and the sentence has been modified in the manuscript.

Point 22: L38: do not = could not

Response 22: Thanks very much for the reviewer’s suggestion. Changes have been made in the manuscript as requested by the reviewer.

Point 23: L46: inverted order of subject and predicate is not convenient for scientific texts

Response 23: Thanks to the reviewer’s suggestion. We have made correction according to the reviewer’s suggestions.

Point 24: L51: When using formal written conventions, a conjunction, by definition, connects two clauses within a sentence. As such, a sentence cannot start with a conjunction when writing formally because there is no initial clause to connect to the second.

Response 24: Thanks very much for the reviewer’s suggestion. We have made correction according to the reviewer’s suggestions.

Point 25: L53: The genus is a taxonomic term, it is not an object which infects arthropod itself, better say “the representatives of the genus infect”

Response 25: Thanks very much for the reviewer’s suggestion. We have made correction according to the reviewer’s suggestions.

Point 26: L54: “currently” doesn’t correspond to the past tense, it’s more likely the present

Response 26: Thanks to the reviewer’s suggestion. The tense has been changed to present tense in the manuscript as requested by the reviewer.

Point 27: L129: After = Then

Response 27: Thanks very much for the reviewer’s suggestion. We have made correction according to the reviewer’s suggestions.

Point 28: L145-146: the predicate is not obvious here

Response 28: Thanks very much for the reviewer’s suggestion. Sentences have been changed in the manuscript as requested by the reviewer.

Point 29: L161: sisterclades = sister clades

Response 29: Thanks to the reviewer’s suggestion. We have made correction according to the reviewer’s suggestions.

Point 30: L271: a mixture of singular and plural forms

Response 30: Thanks very much for the reviewer’s suggestion. We have exchange perithecia to perithecium.

Point 31: L285 and elsewhere: I guess several plants must have more than one trunk

Response 31: Thanks very much for the reviewer’s suggestion. We have made correction according to the reviewer’s suggestions.

Point 32: L355: highly = high

Response 32: Thanks to the reviewer’s suggestion. We have made correction according to the reviewer’s suggestions.

Point 33: L407: materials = agents

Response 33: Thanks to the reviewer’s suggestion. We have made correction according to the reviewer’s suggestions.

Point 34: L410-413: starting two consequent phrases with identical sentences is a poor practice

Response 34: Thanks very much for the reviewer’s suggestion. We have made correction according to the reviewer’s suggestions.

Round 2

Reviewer 2 Report

the comments are given above

see above

Round 3

Reviewer 2 Report

As I already tried to explain, the phylogenetic reconstruction showing overall clustering of the new and previously described taxa, is not a strict indicator of taxonomic allocation, as it may vary depending upon many factors. Sequence similarity 

The paper should include at least a brief summary, if not a detailed analysis, of  sequence similarity\genetic distance between the previously described and the new taxa, so that the clear genetic borders, which are not always evident from the phylograms, are obvious. 

Author Response

Dear Mr. William Zhang

Thank you for your letter and the reviewers’ comments. We have carefully studied the comments, made corrections, and supplemented them in hopes of obtaining approval. The main corrections and responses to the reviewer’s comments are submitted in this paper.

Response to Reviewer 2 Comments

Point 1: As I already tried to explain, the phylogenetic reconstruction showing overall clustering of the new and previously described taxa, is not a strict indicator of taxonomic allocation, as it may vary depending upon many factors. Sequence similarity.

The paper should include at least a brief summary, if not a detailed analysis, of  sequence similarity\genetic distance between the previously described and the new taxa, so that the clear genetic borders, which are not always evident from the phylograms, are obvious.

Response 1: As requested by the reviewers, we have included the calculation of the genetic distance between the previously described and the new taxa.

Thank you. I look forward to hearing from you again.

Best regards.

Sincerely,

Hong Yu

Yunnan Herbal Lab

College of Ecology and Environmental Sciences

Yunnan University 

E-mail: hongyu@ynu.edu.cn